# Onchocerciasis Drug Discovery: In Vitro Evaluation of FDA-Approved Drugs against *Onchocerca gutturosa* in Gambia

**DOI:** 10.3390/pharmaceutics16020210

**Published:** 2024-01-31

**Authors:** Suzanne Gokool, Simon Townson, Andrew Freeman, Jadzia Siemienski-Kleyn, Jakub Zubrzycki, Senyo Tagboto, Marc P. Hübner, Ivan Scandale

**Affiliations:** 1Tropical Parasitic Diseases Unit, Northwick Park Institute for Medical Research, Watford Road, Harrow, London HA1 3UJ, UK; suzanne.gokool@bristol.ac.uk (S.G.); stagboto@uhas.edu.gh (S.T.); 2Institute for Medical Microbiology, Immunology and Parasitology, University Hospital Bonn, Venusberg-Campus 1, 53127 Bonn, Germany; huebner@uni-bonn.de; 3German Center for Infection Research (DZIF), Partner Site Bonn-Cologne, 53127 Bonn, Germany; 4Drugs for Neglected Diseases Initiative, 1202 Geneva, Switzerland; iscandale@dndi.org

**Keywords:** onchocerciasis, drug discovery, anthelmintics, *O. gutturosa*, motility and MTT inhibition, FDA-approved drugs

## Abstract

Onchocerciasis treatment and control relies mainly on the use of ivermectin which has high activity against the microfilarial stage of *Onchocerca volvulus* but limited activity against the long-lived, tissue dwelling adult nematodes. As this neglected tropical disease has now been targeted for elimination, there is an urgent need for new drugs to combat these parasites, ideally with macrofilaricidal activity. In this study, we have examined the anti-*Onchocerca* activity of a range of existing FDA-approved drugs with a view to repurposing, which can lead to rapid and relatively inexpensive development. From the Pharmakon-1600 library, 106 drugs were selected and tested against *O. gutturosa* adult male parasites using a concentration of 1.25 × 10^−5^ M in an in vitro 5-day standard assay to assess motility and viability (using MTT/formazan colorimetry). The findings revealed that 44 drugs produced marginal/moderate activity (50–99% motility and/or MTT reductions) including cefuroxime sodium, methenamine, primaquine phosphate and rivastigmine tartrate, while 23 drugs produced good activity (100% motility reductions and significant MTT reductions), including atovaquone, isradipine, losartan, rifaximin, cefaclor and pyrantel pamoate. Although this study represents only a first step, some of the identified hits indicate there are potential anti-*Onchocerca* drug candidates worthy of further investigation.

## 1. Introduction

Onchocerciasis (river blindness) is caused by the tissue-dwelling filarial worm *Onchocerca volvulus*. The infection is transmitted between humans, by the bite of the blackfly vector of the *Simulium* genus. Circulating microfilariae (mf) accumulate in the skin but in high-intensity infections, the mf can also enter the tissues of the eye. Death of the mf causes the pathologies of pruritus, skin atrophy, skin de-pigmentation, papular rash, eye lesions and blindness in humans [1]. Related to high morbidity is reduced work productivity, which can then lead to social stigmatisation; this poverty promoting nematode infection has been included within the group of neglected tropical diseases (NTDs) [2].

Without an available vaccine, current preventive chemotherapy and elimination of onchocerciasis rely on mass drug administration (MDA) programs (large scale distribution without diagnosis and supervision of health-care staff) that distribute ivermectin (Mectizan^®^, Merck, Rahway, NJ, USA) either on an annual or biannual basis [2]. This drug, belonging to the family of macrocyclic lactones, targets only the mf stage of the parasite by killing mf and exerting embryostatic effects on the adult female worm; that is, mf release from the uterus is temporarily suppressed [3]. Both of these mechanisms have the effect of reducing the skin mf by up to 99% two months after treatment [4]. In order to have an impact on filarial transmission reduction, treatment needs to be administered for the duration of the reproductive life span of the long-lived (up to 15 years [5]), tissue-dwelling adult worms, so program success is constrained by the absence of drugs with macrofilaricidal activity. There are now concerns of developing ivermectin resistance in *Onchocerca* parasites, as has already been reported in nematodes of veterinary importance [6,7,8], and construed in human onchocerciasis [9]. A study in endemic communities exposed to frequent rounds of treatment in Ghana demonstrated that although ivermectin retained microfilarial activity, sub-optimal responses to treatment could be due to the development of resistance by *O. volvulus*, resulting in a decreased effect on the inhibition of mf release [9]. Similarly, in Cameroon, studies on communities given multiple rounds of ivermectin therapy compared to those that were ivermectin naive indicated that continuous exposure to this drug had a reduced embryostatic effect on the female adult worms [10]. The findings from these studies highlight the possible emerging problems of ivermectin resistance and the urgent need for alternative methods of treatment. Recent studies demonstrated that the closely related drug moxidectin (registered for use in humans by the United States Food and Drug Administration (US FDA) in 2018) may offer improved treatment over ivermectin, although no macrofilaricidal effect has been observed so far [11,12].

The Pharmakon 1600 Collection, a US FDA-approved library, consists of 1600 drugs that have reached clinical evaluation and demonstrated biological activity against known targets. By screening a selection of this drug set, the identification of any potential antifilarial candidates could be rapidly repurposed and prove useful in onchocerciasis elimination programs, as many of the drugs represented in the library are available on the market. Humans are the only viable hosts of *O. volvulus* and there are no laboratory models that support the complete life cycle of this parasite; as such, drug discovery for onchocerciasis has to rely in part, on the use of surrogate parasites and animal hosts. Several in vitro and in vivo standard operating protocols for testing drugs against the adult stage of *Onchocerca* have been developed and optimized [13] and references therein. Using a World Health Organisation approved 5-day motility/MTT-based assay with the cattle filarial nematode *Onchocerca gutturosa*, in this study we have assessed the activity of 106 selected drugs with a range of biological activities (see Table 1).

## 2. Materials and Methods

A workflow diagram for the experimental procedure used in this study is depicted in Figure 1.

### 2.1. Parasites—Onchocerca gutturosa

Adult male worms were obtained postmortem from naturally infected, freshly slaughtered cattle in Gambia, West Africa. The material used in this study was purchased from local butchers by the West Africa Livestock Innovation Centre (WALIC), Banjul, Gambia. The adult male worms were dissected from the cattle nuchal ligament connective tissue and transferred to each well of a sterile 24-well (2 mL) plate (Fisher Scientific, Loughborough, UK), and maintained for at least 24 h in culture before use. The culture medium used was Minimum Essential Medium (MEM) with Earl’s Salts and L-Glutamine (Life Technologies Ltd., Loughborough, UK) supplemented with 10% heat inactivated new-born calf serum (Life Technologies Ltd., UK) and 200 units/mL penicillin, 200 µg/mL streptomycin and 0.5 µg/mL amphotericin B (Life Technologies Ltd., UK). Only normally active worms were used for the test and all assays were conducted at 37 °C under an atmosphere of 5% CO_2_ in air.

### 2.2. Origin of Drugs Tested

The drugs of the Pharmakon 1600 Collection were supplied by MicroSource Discovery Systems Inc. (Gaylordsville, CT, USA) as 10 mM DMSO stock solutions in microtubes in a 96-well plate format and stored at −20 °C. The positive control drug, Immiticide^®^ (melarsomine dihydrochloride, Merial, Duluth, GA, USA) was supplied as a dry solid, known amounts of this drug were solubilized in 1 mL DMSO.

### 2.3. In Vitro Drug Activity against O. gutturosa Adult Worms as Described by Townson et al. [13]

The selected Pharmakon drugs with the positive control Immiticide^®^ were screened using a final concentration of 1.25 × 10^−5^ M. At this concentration, the drugs were not toxic to mammalian cells (LLMCK2 monkey kidney cells, evaluated by microscopy). Two trials were performed using different numbers of worms (according to availability) and drugs. Trial 1: Four worms per group were used for the test and positive control drugs, and eight worms for the untreated control group. Only 100 of the selected 106 drugs were tested due to an insufficient number of available worms. Trial 2: This was performed using three separate assays; 23 drugs were tested, of which 17 were retested from Trial 1 (confirmation tests), and the 6 remaining drugs that could not be tested in Trial 1; two worms per group were used for the test and positive control drugs and six worms for the untreated control group. Worm viability was assessed using the following parameters: measurement of mean worm motility scores on a scale of 0 (immotile) to 10 (maximum) every 24 h, terminating at 120 h, using an Olympus inverted microscope. Biochemical evaluation of worm viability using MTT/formazan colorimetry was carried out after the last motility reading (120 h), outlined in Figure 1. Inhibition of formazan formation is correlated with worm damage or death (viability). The results of the test drugs were compared to the untreated controls. The results were calculated as motility reduction (%) and MTT reduction (%) compared to the untreated controls (Microsoft Office Excel, 2010) and designated: good activity, 100% motility and/or MTT reduction; moderate/marginal activity, 50–99% motility and/or MTT reduction; inactive, <50% motility reduction and MTT reduction. The test drug is considered active when the motility and/or MTT reductions of ≥50% is observed by comparison to the untreated controls. Statistical analysis was performed on only the Trial 1 data, using the two-tailed t test for the comparison between the motility and optical density means of the test drug and untreated control (Microsoft Office Excel, 2010) with a significance level of *p* < 0.05.

## 3. Results

Of the 106 selected drugs tested, 39 were inactive (<50% motility reduction and MTT reduction), 44 showed marginal/moderate activity (50–99% motility and/or MTT reduction) and 23 showed good activity (100% motility and/or MTT reduction) after 120 h drug exposure.

The results for the 44 drugs that showed moderate or marginal activity in Trial 1 are shown in Table 2. For the majority of these drugs, the MTT reductions correlated with the motility reductions, indicating that the worms did not only have reduced motility but were permanently damaged. Examination of the data shows that for 13 drugs with the greatest motility reductions of >80%, the MTT reduction was ≥70% (*p*-values < 0.001), with the highest number of hits in the antibacterial (cefuroxime sodium, demeclocycline hydrochloride, methenamine), antiparasitic (amitraz, primaquine phosphate) and neurological (armodafinil, chlorpromazine, dopamine hydrochloride, rivastigmine tartrate) bioactivity groups.

The 23 drugs which demonstrated good activity are shown in Table 3 (includes the chemical structures); 17 of these drugs were retests from Trial 1, and 6 drugs were new tests in Trial 2. All of the drugs produced 100% motility reduction and high levels of MTT reduction, and for most drugs there was a good level of concordance between the results of Trial 1 when compared to Trial 2. The highest number of hits was found in the antibacterial (cefaclor, chlorhexidine dihydrochloride, gramicidin, lasalocid sodium, nitrofurantoin, nitrofurazone, nitroxoline, rifaximin), anti-infective (benzethonium chloride, dequalinium chloride, methylbenzethonium chloride, oxyquinoline hemisulfate) and antiparasitic (atovaquone, hexetidine, homidium bromide, iodoquinol, levamisole hydrochloride, pyrantel pamoate) bioactivity groups.

## 4. Discussion

With the urgent need to identify drugs with potential macrofilaricidal activity against *Onchocerca* parasites, using the strategy of drug repurposing to identify new drugs for the prompt development of therapeutics for the treatment of filariasis is not a new concept; indeed, the drugs currently in use to treat filarial infections, ivermectin, diethylcarbamazine, moxidectin and doxycycline, have all been repurposed from the veterinary or medical fields [14]. Previous studies have screened libraries and drugs for activity against filarial parasites [15] and the Pharmakon 1600 library itself has also been screened for antischistosomal activity [16]. All human-infecting filarial nematodes, with the exception of *L. loa*, carry the endosymbiotic *Wolbachia* bacteria which has been shown to be essential for *O. volvulus* fertility and viability [17,18]. Studies using drugs from diverse libraries in anti-*Wolbachia* screens have revealed promising candidates for further development [19]. Rifampicin (also known as rifampin), an antibiotic used for the treatment of tuberculosis, has been developed for testing in clinical trials based on the effect of high dose, long term exposure using in vitro assays against *O. gutturosa* adult male worms [20] and *Wolbachia* [21]. In addition, in vivo studies have demonstrated more than 90% *Wolbachia* depletion using *Brugia malayi* and *Onchocerca ochengi* models [22]. To test whether the treatment time for onchocerciasis could be reduced, rifampicin in combination with albendazole is currently being investigated in a clinical trial in Cameroon [23]. Emodepside, a veterinary drug licensed for the oral treatment of gastrointestinal nematodes, exhibited high activity in vitro and in vivo against various filarial parasites [24]. A Phase I clinical trial in healthy humans has been completed [25] and recruitment for a Phase II trial in onchocerciasis patients is currently underway [26]. Similarly, oxfendazole is a drug that is used against intestinal helminths in the veterinary field over several decades. In vitro and in vivo animal studies showed that oxfendazole is also active against filarial nematodes [27,28]. Oxfendazole was tested in Phase 1 clinical trials [29] and will be tested in onchocerciasis, loiasis, mansonellosis and *Trichuris trichiura* patients in a Phase 2 clinical basket trial [30]. In this study, we aimed to identify selected existing drugs contained within the FDA-approved Pharmakon 1600 library with the potential to be rapidly developed as macrofilaricides against onchocerciasis.

The results of the standard anthelmintics, contained within the antiparasitic bioactivity group, were as expected and demonstrate that screening of diverse libraries using this 5-day motility/MTT in vitro assay is suitable for identifying new drug candidates with activity against *Onchocerca* parasites. Nevertheless, these types of in vitro assays do not tell us all we need to know about the activity of drugs, since host factors may play an important role. Diethylcarbamazine citrate and flubendazole were inactive in this study and this result corresponds to previous in vitro findings [27,31,32]. However, clinical studies in Mexico demonstrated that flubendazole has high macrofilaricidal activity [33,34] and in vivo laboratory studies confirmed this activity [35]. Recently, the important role of the immune system in supporting the macrofilaricidal efficacy for the related drug oxfendazole was demonstrated in the *Litomosoides sigmodontis* filarial mouse model [28]. Despite the limitation that some candidates may require an intact immune system for efficacy, several promising candidates were identified. In this study, levamisole hydrochloride displayed good activity (100% motility reduction), the parasites were completely immotile after 5 days of exposure (Table 3), and this result is in accordance with previous studies [31], although the MTT result indicated paralysis rather than worm death. Unsurprisingly, pyrantel pamoate (anthelmintic), which is used for roundworm and pinworm infections, showed good activity against *O. gutturosa* parasites in this study; this drug is a depolarising neuromuscular blocking agent which causes paralysis of the worms [36]. Treatment of onchocerciasis patients with pyrantel pamoate showed no notable activity against adult worms of *O. volvulus* [37], possibly due to poor oral uptake or suboptimal dosage/treatment length. Moxidectin showed moderate activity; the motility of the parasites was reduced by 78.6% with a comparable reduction in viability (71.8%) indicating slow killing of the worms in vitro (Table 2). This drug was licensed to treat human onchocerciasis in 2018 [38].

Of the drugs that displayed moderate/marginal activity, cefuroxime sodium, methenamine, primaquine phosphate and rivastigmine tartrate had significant effects on the parasites with motility reductions of >90% and MTT reductions of >70% (*p*-values < 0.001, Table 2). Oral formulations are available for these drugs and therefore they should be considered “drugs of interest” to be further investigated for use against *Onchocerca* infections. Cefuroxime sodium and methenamine are used for the treatment of bacterial infections, primaquine phosphate (antimalarial) is used for the treatment of hypnozoites, the dormant form of *Plasmodium* parasites during malaria tertiana and rivastigmine tartrate has neurological activity in the treatment of dementia associated with Alzheimer’s or Parkinson’s diseases. Rifampin (internationally known as rifampicin) also showed moderate activity (Table 2) and is currently in development for the treatment of onchocerciasis [15]. Such a slow macrofilaricidal efficacy is known for antibiotics that target and eliminate the *Wolbachia* endosymbionts of filariae, such as doxycycline [15]. The semi-synthetic derivative of rifampin, rifaximin, completely reduced the motility of the parasites (Table 3); however, due to poor absorption, rifaximin is only used to treat gastrointestinal infections.

Drugs belonging to novel classes, with available oral formulations, rendered *O. gutturosa* male worms completely immotile (good activity, 100% motility reduction) but the parasites were not dead as indicated by the MTT reductions (Table 3); longer exposure to these drugs, or a longer period in culture following exposure to the drug, may result in parasite death. Isradipine (calcium channel blocker) and losartan (angiotensin receptor blocker) are both used to treat hypertension and cause vasodilation by blocking different receptors. Penfluridol with neurological activity is commonly used as an antipsychotic drug. The anticancer drug, mitoxantrone hydrochloride, is used in the treatment of prostate cancer and leukaemia. Of interest is the activity of atovaquone (antimalarial) against the *O. gutturosa* worms. In addition to its use as prophylaxis and treatment against malaria parasites, it is also used for the treatment of pneumonia caused by fungal infection and some other microbial infections [39]. Several studies have investigated this drug for the treatment of different types of cancers [40]. Also of interest is the antibacterial cefaclor which belongs to the large cephalosporin family of antibiotics; this drug class is structurally related to penicillin and used to treat a wide range of bacterial infections. Cefuroxime sodium, which was moderately/marginally active against the parasites, together with cefaclor, are both second-generation drugs and oral formulations are available for both drugs. We speculate that the active antibacterial drugs tested in this study had a direct effect on worms, with the possibility of an indirect effect by killing *Wolbachia* in the longer term.

In this study we have taken the first step to identify FDA-approved drugs with potential anti-*Onchocerca* activity; some of the identified hits should be further investigated for repurposing. Of lower priority are the drugs designed for topical use and those that can only be administered parenterally. Further investigation of the candidate drugs that displayed promising in vitro activity against *O. gutturosa* adult male parasites was curtailed due to the unexpected, imposed travel and work restrictions with regard to studies in Gambia during the COVID-19 pandemic. Further in vitro trials are required to retest the hits of interest to estimate activity endpoints and EC_50_ values. This data together with the available pharmacokinetic and toxicity profiles can be used to rapidly inform the development of the drugs for further evaluation against *Onchocerca* and other filarial species of medical and veterinary importance. In addition, these hits may provide a good starting point to assess related compounds of interest and for the synthesis of new drugs.

## Figures and Tables

**Figure 1 pharmaceutics-16-00210-f001:**
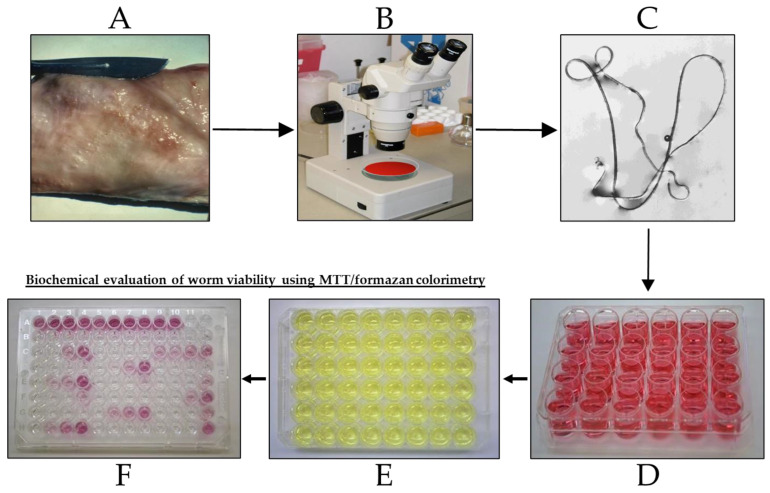
Workflow diagram for the *Onchocerca gutturosa* adult worm in vitro 5-day motility/MTT assay. (**A**) Nuchal ligament connective tissue from naturally infected cattle; (**B**) Tissue in culture medium placed under a dissecting microscope (40× magnification) to isolate worms using fine forceps; (**C**) Isolated worm floating on tissue culture medium, worm length approximately 2 cm; (**D**) Worms were maintained individually in each well of a 24-well plate containing 2 mL culture medium, at 37 °C under an atmosphere of 5% CO_2_ in air; this was replaced after 24 h with culture medium containing the test drug at a final concentration of 1.25 × 10^−5^ M. Worm motility was determined microscopically every 24 h up to 120 h; (**E**) Worm viability was then assessed by the transfer of each worm to the well of a 48-well plate containing 0.5 mg/mL MTT, incubation at 37 °C for 30 min; (**F**) Each worm was transferred to a well in a 96-well plate containing 200 µL dimethylsulfoxide to solubilize the formazan. After 1 h at 23–26 °C, the plate was gently agitated and formazan formation measured using absorbance (490 nm) on an ELISA reader (Dynatech, Willenhall, UK).

**Table 1 pharmaceutics-16-00210-t001:** Selected Pharmakon-1600 drugs (106) for in vitro evaluation against *O. gutturosa* adult male parasites. Bioactivity was categorised by MicroSource Discovery Systems, Inc. (Gaylordsville, CT, USA).

Bioactivity	Drug
Antibacterial	Carbadox; Cefaclor; Cefamandole nafate; Cefoperazone; Cefoxitin sodium; Cefsulodin sodium; Ceftibuten; Cefuroxime sodium; Chlorhexidine dihydrochloride; Chloroxylenol; Demeclocycline hydrochloride; Doxycycline hydrochloride; Furazolidone; Gramicidin; Lasalocid sodium; Merbromin; Methacycline hydrochloride; Methenamine; Minocycline hydrochloride; Nitrofurantoin; Nitroxoline; Ofloxacin; Oxytetracycline; Rifampcin; Rifaximin; Sulfaquinoxaline sodium; Teicoplanin
Anticancer	Azaserine; Bleomycin; Daunorubicin; Doxorubicin; Epirubicin hydrochloride; Isotretinon; Lomustine; Mitoxantrone hydrochloride; Tretinoin
Antihypertensive/ vasodilator	Dipyridamole; Guanethidine; Losartan; Nicardipine hydrochloride; Nicotinyl alcohol tartrate; Nifedipine
Anti-infective	Benzethonium chloride; Broxyquinoline; Dequalinium chloride; Methylbenzethonium chloride; Nitrofurazone; Oxyquinoline hemisulfate; Phenylethyl alcohol; Resorcinol monoacetate
Anti-inflammatory/ antihistamine	Dexamethasone acetate; Doxylamine succinate; Meloxicam sodium; Prednisolone tebutate; Sulfasalazine
Antiparasitic	Amitraz; Atovaquone; Candicidin; Clorsulon; Diethylcarbamazine citrate; Flubendazole; Hexetidine; Homidium bromide; Iodoquinol; Levamisole hydrochloride; Moxidectin; Primaquine phosphate; Pyrantel pamoate;
Antiviral	Oseltamivir phosphate; Valganciclovir hydrochloride
Neurological	Acepromazine maleate; Almotriptan; Ampyzine sulfate; Apomorphine hydrochloride; Armodafinil; Bupropion; Chlorpromazine; Danazol; Desipramine hydrochloride; Dopamine hydrochloride; Isradipine (also antihypertensive/vasodilator); Methsuximide; Methylphenidate hydrochloride; Olanzapine; Oxidopamine hydrochloride; Penfluridol; Rivastigmine tartrate; Selegiline hydrochloride; Zaleplon
Various	Alendronate sodium; Anisindione; Ascorbyl palmitate; Bromhexine hydrochloride; Butacaine; β-Carotene; Clopidogrel sulfate; Dienestrol; Dioxybenzone; Docusate sodium; Fluorescein; Mangafodipir trisodium; Methylergonovine maleate; Propoxycaine hydrochloride; Riboflavin; Sennoside A; Tetrahydrozoline hydrochloride

**Table 2 pharmaceutics-16-00210-t002:** Results of the 44 identified drugs with moderate or marginal activity after 120 h drug exposure. Mot Redn—motility reduction; MTT Redn—MTT reduction.

Drug	Mot Red		MTT Red	
	%	*p*-Value	%	*p*-Value
IMMITICIDE (positive control)	100.00	<0.0001	91.09	<0.0001
Acepromazine Maleate	53.57	<0.0001	57.67	<0.001
Amitraz	89.29	<0.0001	77.23	<0.0001
Ampyzine Sulfate	78.57	<0.0001	76.98	<0.0001
Apomorphine Hydrochloride	78.57	<0.0001	76.49	<0.0001
Armodafinil	82.14	<0.0001	84.65	<0.0001
Ascorbyl Palmitate	57.14	<0.0001	67.08	<0.001
Bleomycin	53.57	<0.0001	53.22	<0.01
Bromhexine Hydrochloride	57.14	<0.0001	56.93	<0.01
Broxyquinoline	82.14	<0.0001	82.18	<0.0001
Candicidin	67.86	<0.0001	72.03	<0.001
Carbadox	60.71	<0.0001	66.83	<0.001
Cefsulodin Sodium	50.00	<0.0001	21.04	0.18
Ceftibuten	75.00	<0.0001	71.53	<0.001
Cefuroxime Sodium	92.86	<0.0001	79.70	<0.0001
Chlorpromazine	82.14	<0.0001	79.46	<0.0001
Clopidogrel Sulfate	28.57	<0.01	53.47	<0.01
Demeclocycline Hydrochloride	82.14	<0.0001	76.24	<0.001
Dexamethasone Acetate	78.57	<0.0001	61.14	<0.001
Dienestrol	78.57	<0.0001	75.74	<0.0001
Docusate Sodium	75.00	<0.0001	66.58	<0.001
Dopamine Hydrochloride	82.14	<0.0001	69.55	<0.001
Doxorubicin	60.71	<0.0001	64.36	<0.001
Doxycycline Hydrochloride	53.57	<0.0001	37.13	<0.05
Epirubicin Hydrochloride	75.00	<0.0001	77.48	<0.0001
Fluorescein	50.00	<0.0001	45.54	<0.01
Guanethidine	78.57	<0.0001	79.95	<0.0001
Mangafodipir Trisodium	89.29	<0.0001	76.49	<0.0001
Methenamine	92.86	<0.0001	78.47	<0.0001
Methsuximide	71.43	<0.0001	66.34	<0.001
Methylergonovine Maleate	57.14	<0.0001	55.45	<0.001
Methylphenidate Hydrochloride	75.00	<0.0001	88.12	<0.0001
Minocycline Hydrochloride	78.57	<0.0001	72.52	<0.001
Moxidectin	78.57	<0.0001	71.78	<0.001
Nicotinyl Alcohol Tartrate	67.86	<0.0001	70.30	<0.001
Nifedipine	92.86	<0.0001	82.92	<0.0001
Prednisolone Tebutate	96.43	<0.0001	82.18	<0.0001
Primaquine Phosphate	92.86	<0.0001	84.16	<0.0001
Propoxycaine Hydrochloride	60.71	<0.0001	60.15	<0.001
Riboflavin	60.71	<0.0001	56.19	<0.01
Rifampin	57.14	<0.0001	56.44	<0.01
Rivastigmine Tartrate	92.86	<0.0001	81.93	<0.001
Sennoside A	53.57	<0.0001	52.23	<0.01
Tetrahydrozoline Hydrochloride	53.57	<0.0001	53.22	<0.01
Valganciclovir Hydrochloride	50.00	<0.0001	53.71	<0.01

**Table 3 pharmaceutics-16-00210-t003:** Results of the 23 drugs identified with good activity after 120 h drug exposure. Mot Redn—motility reduction; MTT Redn—MTT reduction; nd—not determined; *p*-value applies to both motility and MTT reductions.

Drug (Molecular Weight/Bioactivity)	Molecular Structure	Trial 1			Trial 2	
		Mot Redn (%)	MTT Redn (%)	*p*-Value	Mot Redn (%)	MTT Redn (%)
IMMITICIDE, positive control (501.34/Anthelmintic)	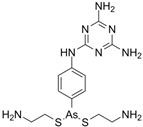	100.00	91.09	<0.0001	100.00	Range 77.19–98.88
ATOVAQUONE (366.85/Antimalarial)	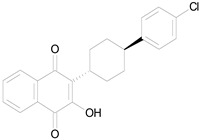	100.00	90.10	<0.0001	100.00	74.42
BENZETHONIUM CHLORIDE (448.09/Antiinfective)	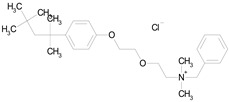	100.00	88.12	<0.0001	100.00	71.91
CHLORHEXIDINE DIHYDROCHLORIDE (578.38/Antibacterial)	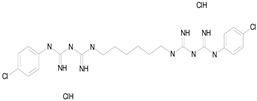	100.00	90.10	<0.0001	100.00	100.00
GRAMICIDIN, gramicidin A shown (1882.34/Antibacterial)	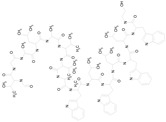	100.00	76.24	<0.0001	100.00	84.27
IODOQUINOL (396.96/Antiamoebic)	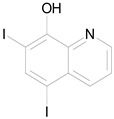	100.00	80.20	<0.0001	100.00	87.64
ISRADIPINE (371.40/Calcium channel blocker)	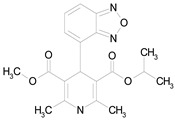	100.00	87.13	<0.0001	100.00	78.95
LASALOCID SODIUM (612.79/Antibacterial)	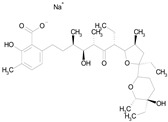	100.00	89.11	<0.0001	73.33	90.70
LEVAMISOLE HYDROCHLORIDE (240.76/Anthelmintic)	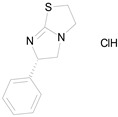	100.00	92.08	<0.0001	100.00	55.81
LOSARTAN (422.92/Antihypertensive)	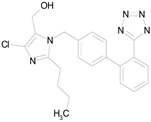	100.00	88.12	<0.0001	100.00	82.56
MELOXICAM SODIUM (373.39/Antiinflammatory)	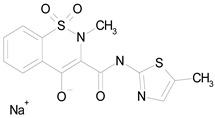	100.00	89.11	<0.0001	100.00	47.67
METHYLBENZETHONIUM CHLORIDE (462.12/Antiinfective)	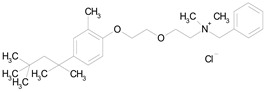	100.00	91.09	<0.0001	100.00	67.44
MITOXANTRONE HYDROCHLORIDE (517.41/Antineoplastic)	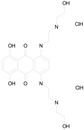	100.00	91.09	<0.0001	100.00	70.93
NITROFURANTOIN (238.16/Antibacterial)	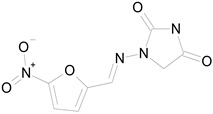	100.00	75.25	<0.0001	100.00	92.70
NITROFURAZONE (198.14/Antibacterial)	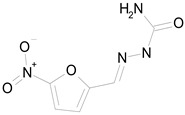	100.00	85.15	<0.0001	100.00	95.51
OXYQUINOLINE HEMISULFATE (243.24/Antiinfective)	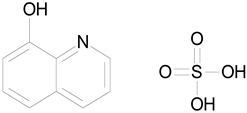	100.00	86.14	<0.0001	100.00	91.57
PYRANTEL PAMOATE (594.69/Anthelmintic)	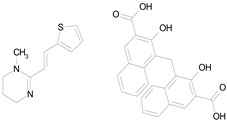	100.00	85.15	<0.0001	100.00	90.45
RIFAXIMIN (785.90/Antibacterial)	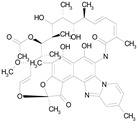	100.00	91.09	<0.0001	100.00	68.60
CEFACLOR (367.81/Antibacterial)	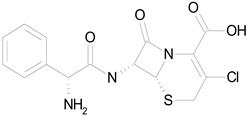	nd	nd	nd	100.00	96.51
DEQUALINIUM CHLORIDE (527.59/Antiinfective)	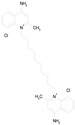	nd	nd	nd	100.00	61.40
HEXETIDINE (339.61/Antifungal)	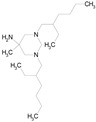	nd	nd	nd	100.00	88.60
HOMIDIUM BROMIDE (394.32/Antiprotozoal)	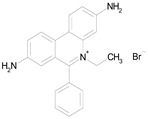	nd	nd	nd	100.00	81.58
NITROXOLINE (190.16/Antibacterial)	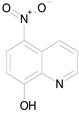	nd	nd	nd	100.00	82.46
PENFLURIDOL (523.98/Antipsychotic)	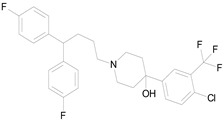	nd	nd	nd	100.00	64.04

## Data Availability

Data are available upon reasonable request.

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
