# Peer review of "Onchocerciasis Drug Discovery: In Vitro Evaluation of FDA-Approved Drugs against Onchocerca gutturosa in Gambia"

_pharmaceutics, 2024, doi:10.3390/pharmaceutics16020210_

Round 1

Reviewer 1 Report

Comments and Suggestions for Authors

The authors assess a collection of 106 approved drugs and examine if they may be repurposed to kill adult O. volvulus, the filarial worm responsible for river blindness. O. volvulus has a complex life cycle, including adults and embryos (microfilariae) in humans, and other development stages in its insect vector (Simulium flies). The current drugs, including ivermectin which is used in mass drug administration programs against river blindness, kill the microfilariae. This helps limiting the spread of infection to other members of the community, since microfilariae are what is transmitted from a person to a fly. However, ivermectin does not kill the adult worms, which can live for many years. This forces MDA programs to be repeated at least once a year for decades, which is far from being optimal. The quest for drugs that can kill adults (macrofilariae), ideally in a single dose, is therefore of the utmost importance. 

Here the authors have identified several drugs that may be promising as macrofilaricides in a first screen.  As the authors have pointed out, this is only a first step, but it is a very important one. 

An excellent manuscript - concise and to the point, containing very important data.  

Author Response

No revisions required.

Reviewer 2 Report

Comments and Suggestions for Authors

This is an important (although pointlessly long) study. I have no specific suggestions.

Author Response

No revisions required.

Reviewer 3 Report

Comments and Suggestions for Authors

Review for the

Manuscript ID: pharmaceutics-2750746

Title: Onchocerciasis Drug Discovery: In vitro evaluation of FDA- Approved

Drugs against Onchocerca gutturosa in Gambia

Authors: Suzanne Gokool, Simon Townson *, Andrew Freeman, Jadzia

Siemienski-Kleyn, Jakub Zubrzycki, Senyo Tagboto, Marc P. Hübner, Ivan

Scandale

Submitted to section: Drug Targeting and Design,

https://www.mdpi.com/journal/pharmaceutics/sections/Drug_Targeting_Design

Emerging Pharmaceutical Therapeutics for Neglected Tropical Diseases

https://www.mdpi.com/journal/pharmaceutics/special_issues/2ZB1WCZUX2

Current manuscript is suitable for the section : Drug Rargeting and Design.

In general, the paper is dedicated to drug re-purposing with the aim to identify the new agents targeting neglected tropical disease (Onchocerciasis or River blindness) among the wide variety of FDA approved drugs (from Pharmacon-1600 library and 106 drugs were selected for this study).

In this preliminary study authors  identified the most perspective candidate drugs (hits) for further more deep and detailed evaluation  against Onchocerca and other filarial species.

This work  is actual, interesting, and can be useful for researchers, currently working in antiparasitic drug and tropical diseases areas.

In general, current  paper is clearly written and well designed, so, reviewer can only need to add some minor suggestions for further  improvement of manuscript quality (listed in attached PDF).

Author Response

The responses to the suggested revisions are shown in blue in the attached document and the legend of Figure 1 has been amended in the manuscript.

Reviewer 4 Report

Comments and Suggestions for Authors

The materials and methods should be more detailed regarding the procedures used (e.g., temperatures and incubation times).

Author Response

Response to the specific suggestion as shown below:

The temperature and incubation times for each step of the O. gutturosa adult worm in vitro 5-day motility/MTT assay have been provided in the legend of the workflow diagram (Figure 1), therefore this information was not repeated in the text of the Materials and Methods section.